# Needle Senescence Affects Fire Behavior in Aleppo Pine (*Pinus halepensis* Mill.) Stands: A Simulation Study

**Rodrigo Balaguer-Romano** [1,*] , **Rubén Díaz-Sierra** [1] , **Javier Madrigal** [2,3] , **Jordi Voltas** [4,5] **and Víctor Resco de Dios** [4,5,6,*]

1   Mathematical and Fluid Physics Department, Faculty of Sciences, Universidad Nacional de Educación a Distancia (UNED), 28040 Madrid, Spain; sierra@ccia.uned.es
2   Department of Forest Dynamics and Management, INIA–CIFOR, Ctra. A Coruña Km 7.5, 28040 Madrid, Spain; incendio@inia.es
3   ETSI Montes, Forestal y del Medio Natural, Universidad Politécnica de Madrid (UPM), Ramiro de Maeztu, 28040 Madrid, Spain
4   Department of Crop and Forest Sciences, Universitat Lleida, 25198 Lleida, Spain; jordi.voltas@udl.cat
5   Oint Research Unit CTFC-AGROTECNIO, Universitat de Lleida, 25198 Lleida, Spain
6   School of Life Science and Engineering, Southwest University of Science and Technology, Mianyang 621010, China
*   Correspondence: rodrigo.balaguer.romano@gmail.com (R.B.-R.); v.rescodedios@gmail.com (V.R.d.D.)

**Abstract:** *Research Highlights:* Pre-programmed cell death in old Aleppo pine needles leads to low moisture contents in the forest canopy in July, the time when fire activity nears its peak in the Western Mediterranean Basin. Here, we show, for the first time, that such needle senescence may increase fire behavior and thus is a potential mechanism explaining why the bulk of the annual burned area in the region occurs in early summer. *Background and Objectives:* The brunt of the fire season in the Western Mediterranean Basin occurs at the beginning of July, when live fuel moisture content is near its maximum. Here, we test whether a potential explanation to this conundrum lies in Aleppo pine needle senescence, a result of pre-programmed cell death in 3-years-old needles, which typically occurs in the weeks preceding the peak in the burned area. Our objective was to simulate the effects of needle senescence on fire behavior. *Materials and Methods:* We simulated the effects of needle senescence on canopy moisture and structure. Fire behavior was simulated across different phenological scenarios and for two highly contrasting Aleppo pine stand structures, a forest, and a shrubland. Wildfire behavior simulations were done with BehavePlus6 across a wide range of wind speeds and of dead fine surface fuel moistures. *Results:* The transition from surface to passive crown fire occurred at lower wind speeds under simulated needle senescence in the forest and in the shrubland. Transitions to active crown fire only occurred in the shrubland under needle senescence. Maximum fire intensity and severity were always recorded in the needle senescence scenario. *Conclusions:* Aleppo pine needle senescence may enhance the probability of crown fire development at the onset of the fire season, and it could partly explain the concentration of fire activity in early July in the Western Mediterranean Basin.

**Keywords:** fire behavior; crown fire; fire modeling; senescence; foliar moisture content; canopy bulk density

## 1. Introduction

Pine-dominated ecosystems are one of the major landscape types in the Mediterranean Basin, where they cover 25% of the forest surface [1]. One of the most abundant and widespread pine species in the Mediterranean Basin is *Pinus halepensis* Mill. (Aleppo pine), which covers 6.8 Mha, at low altitudes (<500 m) and near the coastline [2]. Aleppo pine is a fire-embracer species meaning that it depends, at least partly, on fires for seed release from serotinous cones and consequent regeneration [3,4]. Post-fire regeneration often results in dense thickets that show a high accumulation of ladder fuels leading to vertical fuel continuity [5]. *P. halepensis* shows a low degree of self-pruning, and these forests are thus particularly prone to crown fires. Approximately one-third of the total annual burned area in the Mediterranean Basin occurs in *P. halepensis* stands [6].

There are different types of crown fires, ranging from individual tree torching, active crown fires and, under exceptional circumstance, independent crown fires that become decoupled from surface fuels may also occur [7]. Wildfire in *P. halepensis* stands often show potential for developing active crown fires beyond extinction capacity [8]. The high rate of spread and intensity of crown fires in *P. halepensis* stands, combined with long range spotting are characteristics that pose a serious threat to life and property [9].

In order to understand potential wildfire behavior, mathematical models have been developed to account for the various interacting processes that drive fire behavior [10]. In North America and Europe, different models that link [11,12] surface and crown fire rate of spread predictions with [7,13] crown fire transition and propagation criteria have often been used [14], including BehavePlus (USDA, Missoula, MT, USA) [15], FlamMap (USDA, Missoula, MT, USA) [16] or NEXUS (USDA, Missoula, MT, USA) [17].

In these semi-empirical approaches, the onset of a crown fire is defined by the transition of a wildfire from surface to canopy fuels. This transition occurs when the surface fire intensity attains or exceeds a certain critical surface intensity ($I_0$), which, in turn, is determined by the interaction between foliar moisture content (FMC) and the canopy base height (CBH) [7]:

$$I_0 = (0.01 \, \text{CBH} \, (460 + 25.9 \, \text{FMC}))^{1.5} \tag{1}$$

After the transition from the surface to the canopy layer, a certain canopy bulk density (CBD) is needed to develop and maintain a solid flame front. If this CBD is not reached, the crown fire will passively torch isolated trees (or groups of trees), but it will not spread across the canopy [17]. Consequently, for active crown fire development, a critical minimum spread rate ($R_0$), which depends on CBD, is needed to maintain continuous crowning [12]:

$$R_0 = \frac{3}{\text{CBD}} \tag{2}$$

Characterization of the fuel structure and its relevance for fire behavior has been the topic of much research [18]. Variations in live fuel moisture are often taken into account, although some discussions are still active on its role in fire propagation [19]. However, an aspect that has seldom been considered is the role of pre-programmed needle senescence, despite its potential to increase crown fire intensity and severity [19,20].

Needle lifespan in *P. halepensis* is approximately three years, and three-years-old needles typically become dry and senesce towards the end of June or start of July (Figure 1A). This is immediately before the peak of the fire season in the Western Mediterranean basin, which often occurs in the first half of July [21] (Figure 1B). Consequently, pre-programmed needle senescence (a developmental process that allows nutrient recycling in old leaves before shedding) potentially leads to one-third of the canopy (that is, all 3 years-old leaves) being dry right before the peak fire season [22].

Some studies have addressed the role of FMC on fire behavior [23]. Others have addressed how canopy drying, following bark beetle attacks, for instance, impacts fire behavior [24–26]. However,

to the best of our knowledge, the effects of partial canopy drying after needle senescence on crown flammability have not been quantified so far [19,20,22].

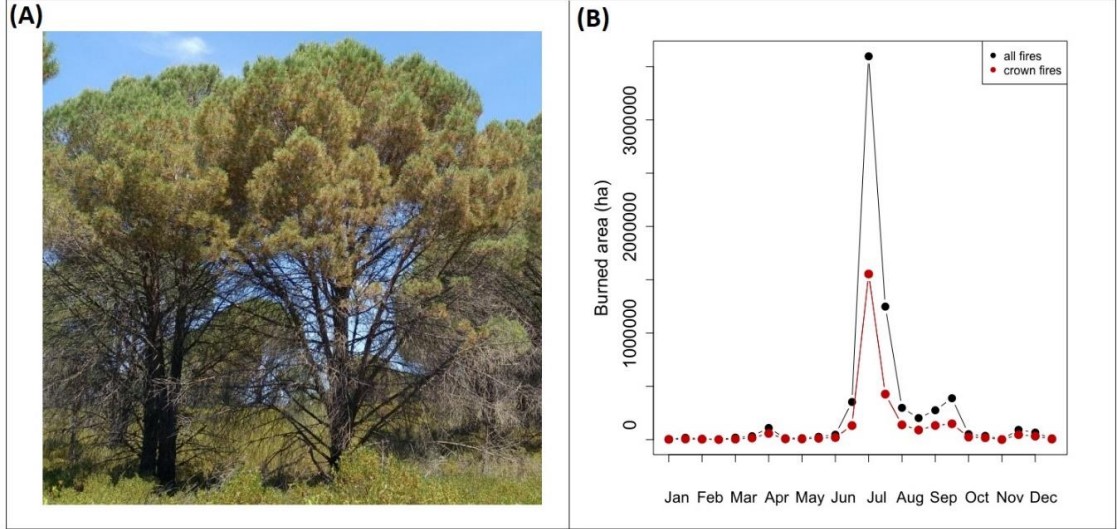

**Figure 1.** (**A**) Needle senescence in *P. halepensis* affecting the old leaves cohort (3-years-old) typically occurs between the end of June or July, and it drastically modifies the moisture of the canopy. Photo by Carles Arteaga. (**B**); Temporal pattern of long-term average (1968–2015) burned area (black, all fires; red, crown fires) in the *Pinus halepensis* forests of the Mediterranean regions of Spain. (Data from the Estadística General de Incendios Forestal provided by the Ministry of Agriculture, Fishing, and Food).

Temporal and spatial coincidence of low foliar moisture content and high canopy bulk density creates optimal conditions to increase the probability of crown fire occurrence as well as their intensity and severity. High-intensity crown fires burn canopies by convection, leading to widespread defoliation and, consequently, plant death. Preprogrammed old needle senescence may thus enhance Aleppo pine mortality rates after wildfires, if it does affect intensity fire behavior [19]. However, this effect only lasts for a few weeks, until leaf dropping [20]. After shedding of senesced needles, the probability of crown fire activity declines as the weighted foliar moisture content increases and the canopy bulk density decreases. Consequently, surface fires may become more intense after needle shedding due to an increase in surface fuel loads, but surface fires seldom reach intensities beyond extinction capacity.

It is currently unknown why the brunt of the fire season occurs in early July in the Western Mediterranean Basin [19]. During this time, FMC in Mediterranean trees, shrubs, and grasses is near its seasonal maximum [27] and fires occurring in late August, under much lower FMC, often burn at lower intensity [19]. Aleppo pine needle senescence could thus offer at least a partial explanation to such conundrum.

Here we seek to quantify the potential effects of needle senescence on fire behavior in *P. halepensis* stands. We simulated four scenarios that recreated the major annual physiological and structural changes in relation to needle senescence (that is, before, during and after leaf senescence and later in the year after the onset of litter decomposition in the autumn). Each of the four simulations was ran for two highly contrasting *P. halepensis* fuel structures (representatives of very high and very low crown fire likelihood) that are dominant in Valencia (E Spain), one of the most fire-prone regions in Mediterranean Spain. We wanted to test the potential effects of needle senescence on crown and surface fire behavior in contrasting stand types, and also to establish its dependence and interactions with wind speed and dead fuel moisture, two well-known drivers of fire behavior. More specifically, we wanted to test: (i) whether needle senescence increases the likelihood of transition from surface to crown fire; (ii) whether once the transition to crown fire has occurred, the likelihood to develop an active crown fire increases with needle senescence in widely contrasting stand structures; (iii) whether

needle senescence increases mortality rates after wildfire activity; and (iv) what is the importance of the effect of needle senescence on crown fire likelihood relative to wind speed and dead fuel moisture.

## 2. Materials and Methods

### 2.1. Senescence Scenarios

Aleppo pine presents a tetracyclic annual shoot elongation process. Once senescence is active (end of June-beginning of July) needles have developed two thirds of the total annual elongation in current year shoots [28]. Thus, considering a three-year needle life span, pre-programmed senescence leads to 1/3.6th, or 28%, of the dried canopy (or dead mass fraction, $f_d$) if all 3 years old needles senesce at once. To simulate annual canopy physiological and structural changes caused by needle senescence, four phenological scenarios were created. The first one, scenario-A (Table 1), represents spring leaf sprout. At this time there is an increase in canopy bulk density, canopy cover and foliar moisture content. Scenario-B (Table 1) represents the time of needle senescence, when about 28% of the canopy is composed of dead matter at the beginning of July. To introduce these changes in FMC, canopy live matter moisture ($M_l$) and canopy dead matter moisture ($M_d$) were weighed ($M_w$) considering $f_d$ as in [29]:

$$\text{FMC} = M_w = f_d M_d + (1 - f_d)M_l \tag{3}$$

**Table 1.** Parameters values in shrub and forest fuel types for each scenario: A, before senescence; B, during senescence; C, after shedding; D, in autumn.

| Forest (TU-3) | A | B | C | D |
|---|---|---|---|---|
| Canopy Cover (%) | 35 | 35 | 35 | 35 |
| Canopy Height (m) | 8 | 8 | 8 | 8 |
| Canopy Base Height (m) | 1.5 | 1.5 | 1.5 | 1.5 |
| Canopy Bulk Density (kg/m$^3$) | 0.15 | 0.15 | 0.1 | 0.1 |
| Fine Fuel Load (t/ha) | 2.5 | 2.5 | 3 | 2.5 |
| 1-h Dead Surface Fuel Moisture (%) | 6 | 5 | 5 | 9 |
| 10-h Dead Surface Fuel Moisture (%) | 7 | 6 | 6 | 10 |
| 100-h Dead Surface Fuel Moisture (%) | 8 | 7 | 7 | 11 |
| Foliar Moisture Content (%) | 105 | 74 | 100 | 100 |
| **Shrub (SH-9)** | **A** | **B** | **C** | **D** |
| Canopy Cover (%) | 100 | 100 | 100 | 100 |
| Canopy Height (m) | 5 | 5 | 5 | 5 |
| Canopy Base Height (m) | 1 | 1 | 1 | 1 |
| Canopy Bulk Density (kg/m$^3$) | 0.22 | 0.22 | 0.15 | 0.15 |
| Fine Fuel Load (t/ha) | 10 | 10 | 10.7 | 10 |
| 1-h Dead Surface Fuel Moisture (%) | 6 | 5 | 5 | 9 |
| 10-h Dead Surface Fuel Moisture (%) | 7 | 6 | 6 | 10 |
| 100-h Dead Surface Fuel Moisture (%) | 8 | 7 | 7 | 11 |
| Foliar Moisture Content (%) | 105 | 74 | 100 | 100 |

Scenario-C simulates the time when needles have been shed, which reduce canopy bulk density. The reduction of dry needles in the crown increases weighted foliar moisture content but needle shedding increases surface fine fuel loads. Finally, scenario-D (Table 1) corresponds to autumn and winter periods when surface fine fuel load decreases due to litter decomposition.

### 2.2. Stand Structures and Fuel Features

Forest structure and fuel loads play a critical role in fire behavior and crown fire susceptibility. We obtained fuel structure data from the fuel models developed by the Fire Service in Valencia, Spain [30]. The Valencian fuel model catalogue adapts the models from Scott and Burgan [31] to E

Spain conditions. We used models SH-9 (shrubland from now on; Tables 1 and 2) and TU-3 (forest from now on; Tables 1 and 2). We will refer to SH-9 as a shrub fuel type, in the sense that it is short stature vegetation, but we note that it has two separate fuel layers (canopy fuels begin at 1 m above ground). It represents stands with a low proportion of large trees, extremely high tree density, and horizontal fuel continuity. In contrast, TU-3 is a forest fuel type representing stands with two separated layers, high proportion of large trees, moderate tree density and moderate to low vertical and horizontal fuel continuity. Both models are considered as dynamic fuels, thus live herbaceous fuels become dead depending on their moisture content [31]. For initial model simulations, dead fuel moisture for scenarios A, B, and C were established according to the low moisture values recorded after heat wave periods [32,33]. As scenario-D represents autumn, dead fuel moisture values are higher due to more benign conditions. We obtained $M_l$ from [34] and $M_d$ from [33]. Additionally, in order to understand the effect of leaf senescence relative to other drivers of fire behavior, we conducted a sensitivity analysis on how different values of 1-h dead surface fuel moisture affected fire behavior. Canopy bulk density, canopy height, and canopy base height were established according to [35]. Changes in canopy bulk density were established considering a reduction of 28% among scenarios before and after senescence, as previously argued. Canopy base height values were considered stable among scenarios because the differences in height between 3 and 2 years-old needles are negligible (<10 cm) for the purpose of these simulations.

**Table 2.** Fuel models SH-9 (shrub) and TU-3 (Forest) parameters values.

| Fuel Parameters | Fuel Model TU-3 | Fuel Model SH-9 |
|---|---|---|
| 1-h Dead Fuel Load | 2.5 t/ha | 10 t/ha |
| 10-h Dead Fuel Load | 0.34 t/ha | 5.5 t/ha |
| 100-h Dead Fuel Load | 0.56 t/ha | 0 t/ha |
| Live Herbaceous Fuel Load | 1.5 t/ha | 3.5 t/ha |
| Live Woody Fuel Load | 2.5 t/ha | 16 t/ha |
| 1-h SAV Ratio | 59.05 cm$^2$/cm$^3$ | 24.60 cm$^2$/cm$^3$ |
| Live Herbaceous SAV Ratio | 52.49 cm$^2$/cm$^3$ | 59.05 cm$^2$/cm$^3$ |
| Live Woody SAV Ratio | 45.93 cm$^2$/cm$^3$ | 49.21 cm$^2$/cm$^3$ |
| Fuel Bed Depth | 40 cm | 134 cm |
| Dead Fuel Moisture of Extinction | 30% | 40% |
| Dead Fuel Heat Content | 18,622.3 kJ/kg | 18,622.3 kJ/kg |
| Live Fuel Heat Content | 18,622.3 kJ/kg | 18,622.3 kJ/kg |

### 2.3. Fire Behavior Modelization

Wildland fire behavior simulation was done using BehavePlus6 [15] and crown fire was calculated using Scott and Reinhardt [17] as input option. The input values used in each stand type and each scenario are detailed in Tables 1 and 2. Slope steepness was set to 0% and 10 m open wind speed was established in a range from 0 to 30 km/h. Final figures were created using R.3.6.1. (Lucent Technologies, Murray Hill, NJ, USA) [36]. Assessment of fire severity were performed using the lethal thresholds (LD) developed by [19]. Thus, a crown fraction burned (CFB) between 0.4–0.8 eliminates 50% of the population (LD$_{50}$), and CFB higher than 0.8–0.9 completely eliminates the population (LD$_{100}$). When CFB remains below 0.4 CFB mortality is negligible (LD$_0$) [19].

### 2.4. Dead Mass Fraction Sensitivity Analysis

We also conducted a sensitivity analysis to assess how a varying proportion of $f_d$ affected the transition ratio from a surface to crown layer. This is important because, assuming that the biomass of each cohort is constant, our previously estimated 28% of $f_d$ would constitute a maximum potential value: needle senescence may start earlier in the year such that different values of $f_d$ may occur when the fire season starts. Surface fire intensity was established from the mean surface intensity across scenarios with an intermediate wind speed of 15 km/h.

## 3. Results

We observed that maximum fire intensity and severity occurred in scenario-B under all wind speeds and fuel types (Table 3). Fire intensity and severity values were higher in the shrub than in the forest fuel model. The highest estimated value of Rate of Spread (ROS) in scenario-B for the forest fuel type was 14.6 m/min at a wind speed of 30 km/h. This value was between 2 and 3 times higher than the peak ROS in the other scenarios (Figure 2A). In the shrub fuel type, the highest ROS was 17.7 m/min, a value that was also reached in scenario-B with a wind speed of 30 km/h. ROS in scenario-B in the shrub fuel type was at least 1.4 times higher than in other scenarios (Figure 3A). The highest fire line intensity reached in scenario-B was 5924 kW/m in the forest stand and 17,179 kW/m in the shrub stand. Peak fire line intensity in scenario-B was 2–3 times higher in the forest fuel type and 1.5 times higher in the shrub fuel type compared to other scenarios (Table 3). The highest flame length occurred in scenario-B and took values of 8.7 m in the forest stand and 17.7 m in the shrub stand. Flame length remained between 2–3.3 m for the forest stand and between 10.1–14.4 m in the shrub stand in the other three scenarios (Table 3).

**Table 3.** Simulated Rate of Spread (m/min), Fire Line Intensity (kW/m), Flame Length (m) and Crown Fraction Burned for each scenario (A, B, C, D) under four 10 m open wind speeds (0, 10, 20, 30 km/h).

| FOREST (TU-3) | Wind Speed (km/h) | A | B | C | D |
|---|---|---|---|---|---|
| Rate of Spread (m/min) | 0 | 0.3 | 0.5 | 0.4 | 0.3 |
| | 10 | 0.9 | 1.2 | 1.1 | 0.9 |
| | 20 | 1.7 | 5.1 | 2.6 | 1.7 |
| | 30 | 5.8 | 14.6 | 6.9 | 3.9 |
| Fire Line Intensity (kW/m) | 0 | 48 | 74 | 69 | 45 |
| | 10 | 130 | 200 | 183 | 121 |
| | 20 | 259 | 1384 | 462 | 240 |
| | 30 | 1393 | 5924 | 1585 | 653 |
| Flame Length (m) | 0 | 0.5 | 0.6 | 0.5 | 0.4 |
| | 10 | 0.7 | 0.9 | 0.9 | 0.7 |
| | 20 | 1 | 3.3 | 1.6 | 1 |
| | 30 | 3.3 | 8.7 | 3.6 | 2 |
| Crown Fraction Burned | 0 | 0 | 0 | 0 | 0 |
| | 10 | 0 | 0 | 0 | 0 |
| | 20 | 0 | 0.35 | 0.06 | 0 |
| | 30 | 0.30 | 0.81 | 0.32 | 0.13 |
| SHRUB (SH-9) | Wind Speed (km/h) | A | B | C | D |
| Rate of Spread (m/min) | 0 | 0.7 | 1 | 0.8 | 0.7 |
| | 10 | 2.1 | 3.1 | 2.2 | 1.8 |
| | 20 | 5.7 | 8.6 | 5.5 | 4.4 |
| | 30 | 12.6 | 17.7 | 11.6 | 9.1 |
| Fire Line Intensity (kW/m) | 0 | 560 | 765 | 586 | 490 |
| | 10 | 1752 | 2615 | 1679 | 1330 |
| | 20 | 5208 | 8228 | 4510 | 3402 |
| | 30 | 12,562 | 17,179 | 10,074 | 7372 |
| Flame Length (m) | 0 | 1.8 | 2.2 | 1.9 | 1.7 |
| | 10 | 3.9 | 5.1 | 3.8 | 3.2 |
| | 20 | 8.0 | 10.9 | 7.3 | 6 |
| | 30 | 14.4 | 17.7 | 12.4 | 10.1 |
| Crown Fraction Burned | 0 | 0.13 | 0.19 | 0.1 | 0.08 |
| | 10 | 0.34 | 0.44 | 0.27 | 0.23 |
| | 20 | 0.63 | 0.79 | 0.49 | 0.43 |
| | 30 | 0.95 | 1 | 0.75 | 0.65 |

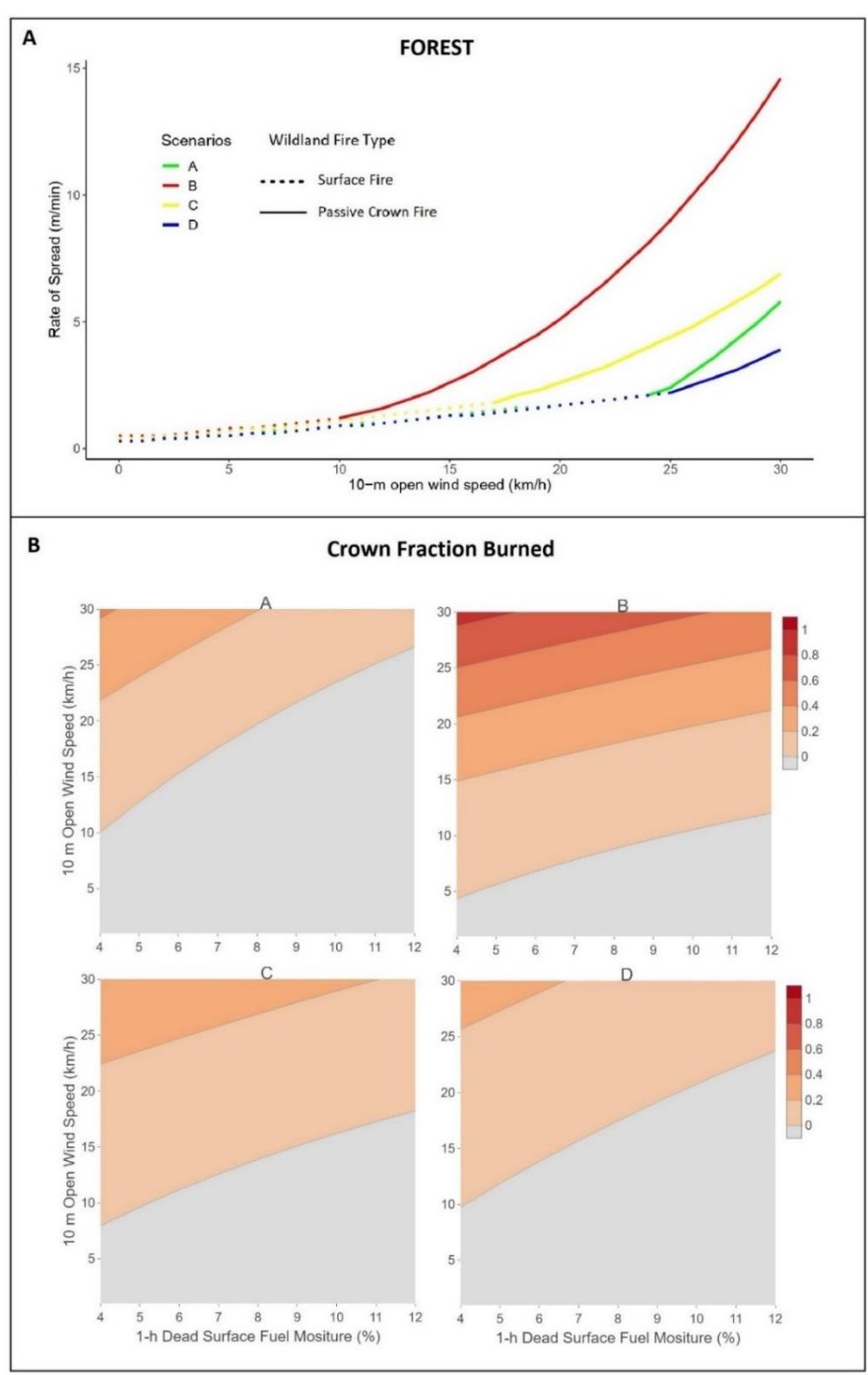

**Figure 2.** (**A**) Rate of Spread (m/min) in each scenario as a function of 10 m open wind speed in forest stands (TU-3 fuel model type). Dotted lines refer to surface fires, solid lines to passive crown fires. (**B**) Crown Fraction Burned values as a function of 10 m open wind speed (km/h) and 1-h dead surface fuel moisture (%) for each scenario: (**A**), before senescence; (**B**), during senescence; (**C**), after shedding; (**D**), in autumn.

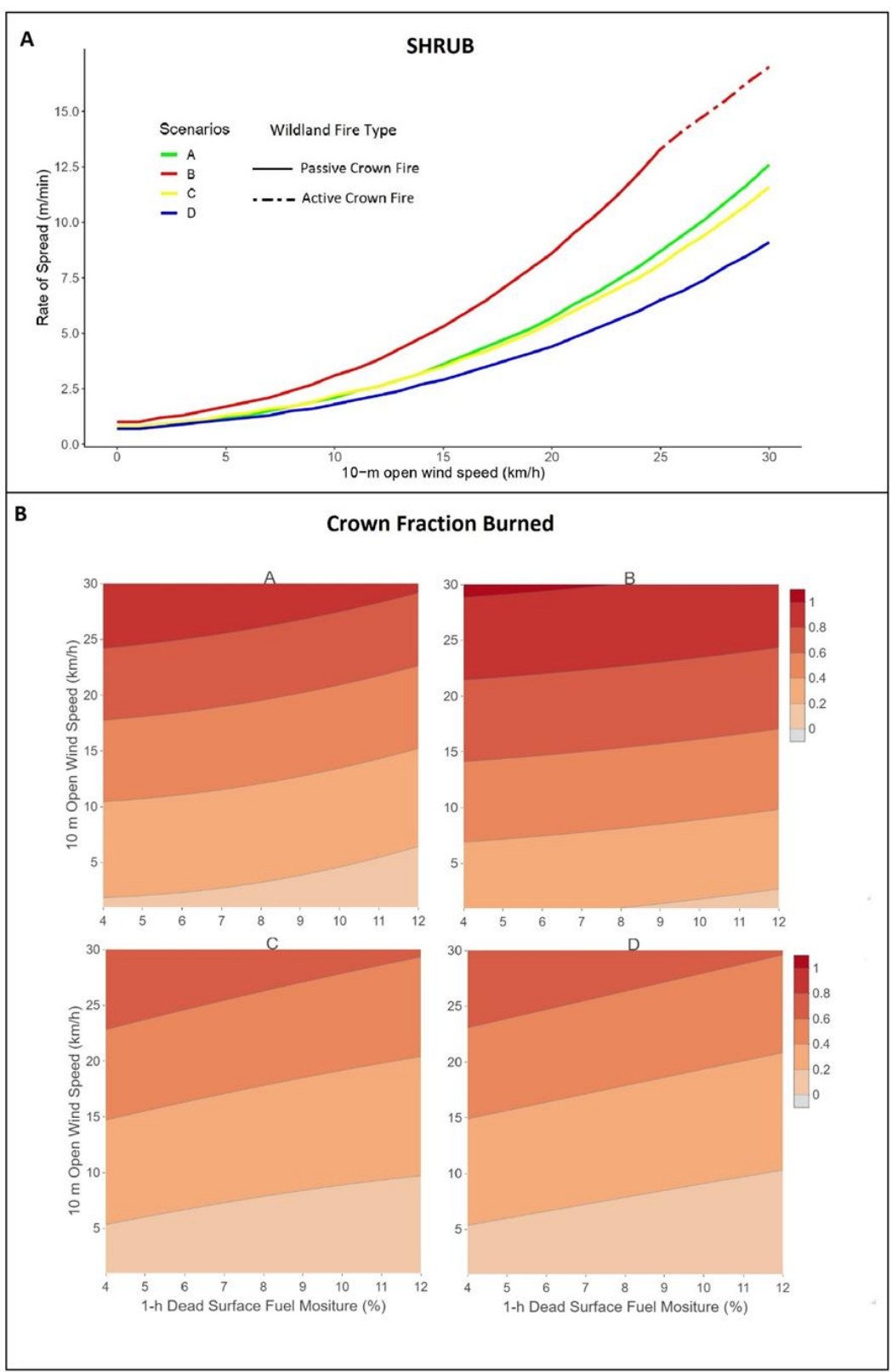

**Figure 3.** (**A**) Rate of spread (m/min) in each scenario as a function of 10 m open wind speed in the shrub stand (SH-9 model type). Solid lines to passive crown fires and dot-dash lines to active crown fires. (**B**) Crown Fraction Burned as a function of 10 m open wind speed (km/h) and 1-h dead surface fuel moisture (%) for each scenario: (**A**), before senescence; (**B**), during senescence; (**C**), after shedding; (**D**), in autumn.

The transition from surface to crown fire in the forest stand occurred with wind speeds higher than 11 km/h in scenario-B. For scenarios A, C and D, the wind speed thresholds necessary for crown fire development were 25, 18 and 26 km/h, respectively. However, it is important to note that we only observed a transition to passive crown fire development, not to active crown fires, in the forest fuel model TU-3. In the shrub fuel model SH-9, passive crown fires developed under all wind speed conditions. Active crown fire only developed in scenario-B, when wind speeds were larger than 25 km/h.

Regarding fire severity, crown fraction burned (CFB) values were always higher in scenario-B for both fuel types and under all wind speed conditions (Figures 2B and 3B). The relative effect of fuel type on CFB was higher in the forest stand than in the shrub stand since maximum CFB was six times larger in scenario-B (0.81) than in scenario-D (0.13). Importantly, the effect on CFB varied markedly with the moisture content of 1-h dead surface fuels. For instance, in the forest, a CBD leading to ($LD_{100}$) in the scenario-B occurred either under a wind speed of 25 km/h and a 1 h dead surface fuel moisture of 4% or with a wind speed of 30 km/h and 1-h dead surface fuel moisture of 10%. $LD_{50}$ was similarly reached with wind speeds above 15 km/h under minimum 1 h dead surface fuel moisture (4%). In the remaining forest scenarios (scenarios A, C, and D), increasing wind speed and lowering 1-h dead surface fuel moisture led to increases in CFB, but they always remained below $LD_{50}$.

In shrublands (Figure 3B), at least some crown damage was recorded in all scenarios under any wind speed and 1-h dead surface fuel moisture conditions. CFB values ranged from 1 in scenario-B to 0.65 in scenario-D under the highest wind speed, indicating important differences depending on fuel phenology. Regarding lethal thresholds (LD), $LD_{50}$ was reached in scenario-B, under a wind speed of 12 km/h when 1-h dead surface fuel moisture was at 12%, or under 8 km/h when 1-h dead surface fuel moisture was at 4%. Further increases in wind speed in this scenario would lead to $LD_{100}$. In the other scenarios, $LD_{50}$ was recorded under an intermediate wind speed of 20 km/h and under critical wind speed conditions (30 km/h), $LD_{100}$ also occurred in scenario A.

Finally, the sensitivity analysis on the effect of a varying $f_d$ on the transition ratio was only performed in forest stands as critical transition to crown fires always occurred in the shrub fuel under any wind speed. Our simulations indicated that the critical surface intensity to crown fire transition under a wind speed of 15 km/h occurred with a minimum fraction of 0.17 of the canopy composed of dead foliar fuels (Figure 4).

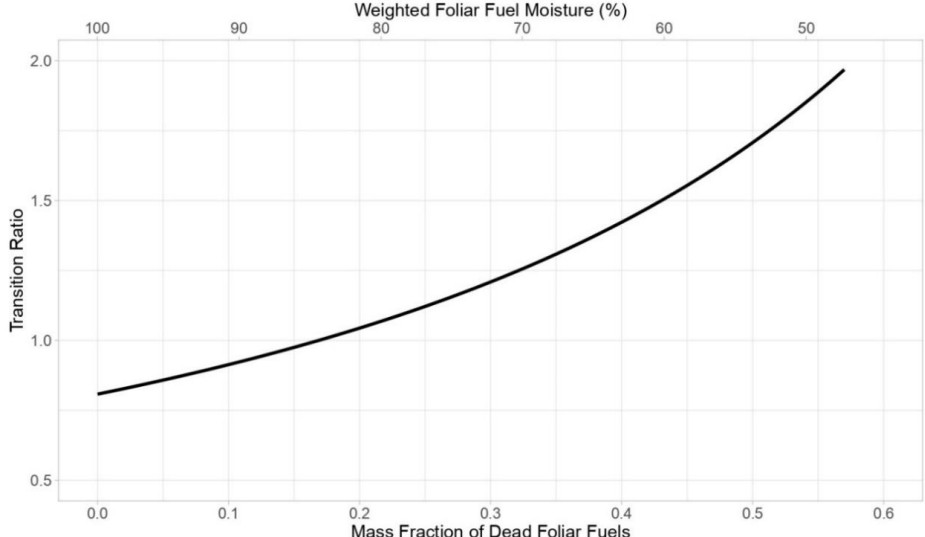

**Figure 4.** Sensitivity analysis on the effects of a varying mass fraction of dead foliar fuels ($f_d$) and associated weighted foliar moisture on the Transition Ratio from a surface fire to the canopy layer on forest stands. Fire transition occurs when the transition ratio between the surface fire intensity (250 kW/m) and critical surface intensity ($I_0$, Equation (1)) becomes equal or higher than 1.

## 4. Discussion

Our results suggest that Aleppo pine needle senescence significantly affects potential crown fire behavior. Simulations showed important differences in wildfire intensity and severity due to the physiological and structural changes caused by needle drying and shedding. However, the effect of needle senescence on fire behavior differed depending on fuel type and its interaction with wind speeds and 1-h dead surface fuel moisture. In other words, needle senescence by itself does not lead to active crown fire, but its presence lowers the critical wind speeds and 1-h dead fuels moisture values necessary to reach such transition point.

We observed stronger crown fire activity under scenario-B in both stand types (Figures 2 and 3). This scenario represents the process of needle senescence leading to a few-weeks period typically occurring towards the end of June or beginning of July [22], during which about one third of pine stand canopy is composed of dry needles (Figure 1A). Spatial and temporal coincidence of low foliar moisture content and high canopy bulk density favors the development of more intense and severe crown fires at lower wind speed conditions, particularly for the shrub fuel type, where active crown fires may develop only under needle senescence. These results indicate that needle senescence could be a contributing factor to increasing fire intensity in Aleppo pine stands. Consequently, this mechanism could partly explain why the peak in the burned area observed in the Western Mediterranean basin, where fires predominantly affect *P. halepensis*, occurs in early July (Figure 1B).

We also observed that the relative effect of needle senescence was more noticeable in the forest fuel model than in the shrub fuel model. This is likely due to the fact that baseline flammability in shrublands is already very high: this fuel type presents a lower canopy base height, which reduces, to some extent, the dependence of critical transitions to crown fire on foliar moisture (Equation (1)). Increasing needle flammability in the shrubland stand would thus have, comparatively speaking, a smaller relative effect for extreme fire behavior than on the forest stand. In fact, crown fires would develop under any wind speed and canopy moisture in shrublands (Figure 3A). However, needle senescence did increase the probability of active crown fires. That is, the development of active fires in the shrubland stand only occurred under canopy senescence. These differences observed between fire behavior in shrub and forest stands are consistent with other studies [5,37,38].

Needle senescence may influence crown fire behavior in at least two ways: affecting FMC and CBD. In our forest stand simulation, we recorded that the wind speed necessary for crown fire development decreased from 25 km/h to 10 km/h between scenarios A and B (Figure 1A) because of decreasing foliar moisture from 105% to 74% (Table 1). A lower FMC reduces the influx of energy required to start the ignition, because a smaller amount of water needs to be evaporated. Needle senescence may thus enhance crown fire development, by reducing foliar moisture content and hence the critical surface intensity threshold value at which surface fires become crown fires. Furthermore, as we observed in the sensitivity analysis (Figure 4), the critical surface intensity to cause the transition from a surface fire to the canopy layer occurred as the dead foliar fractions increased over 17%.

We need to acknowledge that the actual role of FMC in affecting the fire rate of spread is currently being discussed. Some authors argue that the role of FMC is exaggerated in fire behavior models because the high convective and radiative fluxes produced by the flame are several orders of magnitude higher than the energy required to dry the fuel, which would render FMC inconsequent [23]. However, other studies consider that the effect of FMC as a driver of fire spread has actually been underestimated [29,39]. Furthermore, empirical evidence across many biomes support that increases in burnt area occur under decreasing FMC [40–43]. A full discussion on this issue would be out of scope, and the reader is referred to a recent review of this issue for more details [19].

The effect of needle senescence on fire behavior was dependent on 1-h dead surface fuel moisture. As we observed in Figures 2B and 3B, senescence effects interact with variation in 1-h dead surface fuel moisture such that critical CFB values were reached in the senescence scenarios under low 1-h dead surface fuel moisture values. As previously stated, fire behavior is more affected, in relative terms, by the structural and physiological effects caused by needle senescence in forest stands compared to

shrublands. Simulations showed that lethal thresholds varied from $LD_0$, which indicates negligible mortality in all forest scenarios, to $LD_{100}$, which represents the death of the entire population in scenario-B under a wind speed of 30 km/h (Table 3). These changes in tree mortality rates among scenarios were also noticeable in shrublands, where simulations showed that $LD_{100}$ occurred in scenario-B after wind speeds as low as 21 km/h under low 1-h dead fuel moisture values. In the other shrubland scenarios, $LD_{100}$ only occurred in scenario-A, under a critical wind speed condition of 30 km/h. Therefore, while needle senescence is, by itself, not enough to reach critical fire severity thresholds, it lowers the need for critical wind speeds and 1-h dead fuel moisture values necessary to reach $LD_{50}$ or $LD_{100}$.

We recognize that a problem with our study is the way in which the effects of needle senescence on FMC were inputted into the model. We used a weighted average of FMC whereas, in reality, senesced leaves may form a layer of fuel that is effectively independent of live FMC. Future research should concentrate on building more realistic descriptions of needle arrangement such that fuel moisture within a whorl can change with time. We conducted additional simulations considering only the CBD of dead canopy fuels, but the resulting CBD (0.05 kg/m$^3$ for forests and 0.07 kg/m$^3$ for shrublands) was not high enough to produce canopy fires (data not shown).

Another problem with our study lies on the limitations of fire modeling. Considering the complex dynamics behind wildland fires processes, fire models are very simplified, and this could lead to misleading predictions. Furthermore, considering climate change, it is difficult to predict extreme fire conditions accurately. There is some anecdotal evidence that needle senescence enhances crown flammability (M. Castellnou pers. comm.), but further work should confirm experimentally that needle senescence does enhance canopy flammability.

An important yet unresolved aspect is whether needle senescence serves an evolutionary role. It has been reported that pre-programmed needle senescence in the oldest cohort, at least in some temperate and boreal conifers, increases as new leaves develop [44]. This would be a mechanism to recycle nutrients from old leaves into new, developing leaves. In our case, needle senescence co-occurs with the flush of current-year growth, and it could thus serve to support new needle growth. However, needle senescence also occurs as summer drought stress is starting to be important. Consequently, needle senescence could also serve as a water-saving mechanism that decreases transpirational area, at the expense of a transient increase in flammability [22]. However, as climate change intensifies summer drought and wildfire activity, needle senescence could turn maladaptive by enhancing crown fire likelihood. Further efforts towards quantifying the phenology of needle senescence and understanding its underlying drivers should be at the forefront of our research efforts.

We have shown that not considering needle senescence can lead to misleading predictions on fire risk, potentially misestimating wildfire behavior in Aleppo pine stands and this could potentially lead to the application of suboptimal forest and fire management activities. While simulations are routinely performed in order to decide forest management and fire prevention operations, these simulations could incorporate the role of needle senescence because it significantly lowers the threshold for catastrophic fire behavior. To date, needle senescence effects may be underrated in fire behavior simulations due to the relatively short period that it represents each year. However, they occur at a critical time of the year and, as such, its cascading effects on fire behavior may be rather important, as we have anticipated in this work.

An increased probability of extreme events has been forecasted for the next decades as a result of global change. According to predictions, fire seasons may be longer and drier, thereby producing more intense and severe wildfires [19]. Changes in fire regimes represent a challenge to fire-prone species and ecosystems. Aleppo pine post-fire regeneration strategy can be hard-pressed if wildfires return intervals become shorter than the time needed for trees to reach sexual maturity or to produce enough serotinous cones [45]. Also, extremely high wildfire intensity can damage serotinous seeds causing the decline of seedling recruitment and leading to populations collapse [22]. We can thus expect important changes in ecosystem structure in the coming decades, which would have important

interactions with changes in the fire regime. Furthermore, it would be relevant to simulate Aleppo pine-woods responses to predicted future climate conditions for the different scenarios tested in this study. A better understanding of pyrophysiology should, therefore, be at the forefront of our research.

## 5. Conclusions

We have shown evidence, for the first time to our knowledge, of enhanced crown fire behavior in Aleppo pine driven by needle senescence through altered canopy structure and foliage in a period that is coincidental with the brunt of the fire season. Regarding our initial questions, changes in physiological and structural conditions following senescence enhance the probability of more intense and severe crown fires development and concentrate extreme tree mortality rates in senescence periods. Furthermore, in a fuel type with enough canopy bulk density, senescence effects may lead to development of active crown fires. Finally, it is important to consider that senescence, by itself, may not be enough to lead to extreme fire behavior. That is, needle senescence should be viewed as a contributing factor that may favor extreme fire behavior when environmental conditions (e.g., high wind speed) and 1-h dead surface fuel moisture are also at critical levels. We argue for further research to better understand and quantify the drivers of needle senescence and its effects on fire behavior in the field. A lack of consideration of this phenomenon in crown fire modeling systems may provide incomplete predictions leading to the application of unsatisfactory forest and fire management activities.

**Author Contributions:** Conceptualization, V.R.d.D., R.D.-S., J.M. and R.B.-R.; methodology, V.R.d.D., R.D.-S. and R.B.-R.; software, R.B.-R.; writing—original draft preparation, R.B.-R.; writing—review and editing, V.R.d.D., R.D.-S., J.M., J.V. and R.B.-R. All authors have read and agreed to the published version of the manuscript.

**Funding:** We acknowledge funding from the Natural Science Foundation in China (31850410483), the talent proposals from Sichuan Province (2020JDRC0065), the Southwest University of Science and Technology talents fund and the Spanish MICINN (AGL2015-69151-R). R.B.-R. acknowledges the Community of Madrid for the predoctoral contract PEJD-2019-PRE/AMB-15644 funded by the Youth Employment Initiative (YEI).

**Acknowledgments:** We appreciate useful discussion with JL Soriano, MA Botella and the Forest Fire Unit in VAERSA.

**Conflicts of Interest:** The authors declare no conflict of interest.

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
