# Peer review of "Needle Senescence Affects Fire Behavior in Aleppo Pine (Pinus halepensis Mill.) Stands: A Simulation Study"

_forests, doi:10.3390/f11101054_

Round 1

Reviewer 1 Report

The simulation of the effects of needle senescence on fire behavior is very important for prediction of enhance the probability of crown fire development at the onset of the fire season. In this paper, the authors presented interesting results. But there are questions about modeling the process of transition of a surface fire to a crown and propagation of crown fire. It is known that the canopy bulk density in the forest canopy affects the distribution of the velocity field and its decrease affects the resistance of the forest canopy. The question arises whether the authors took this fact into account when applying their mathematical models in the calculations. In the paper the authors provide data that with Aleppo pine needle senescence, the canopy bulk density significantly decreases under different scenarios.

Reviewer 2 Report

Review: “Needle senescence affects fire behavior in Aleppo pine (Pinus halepensis Mill.) stands: a simulation study.” This study uses fire behavior simulations to evaluate the impact of Aleppo pine needle senescence on fire intensity in characteristic Mediterranean forest and shrub stands. The paper is well-written and the results are presented succinctly and clearly. The paper evaluates metrics that are appropriate to the stated research question and clearly demonstrates the impact of needle senescence and corresponding changes in canopy fuel moisture to explain the timing of high-intensity fires in this system. The only additional information I think would be useful to include would be references to real-world examples where the difference between fire behavior in shrub vs forest stands has been observed, in order to bolster the argument. If this has not been observed, it would be important to discuss the reasons why. Relatedly, there should be a more in-depth discussion of the limitations of fire modeling (namely, these are very simplified models of complex dynamics, and more and more as the climate changes, it is difficult to predict extreme fire conditions accurately). Presenting real-world examples from these systems would help support the paper's conclusions.

Introduction: - Good set up for research question, appropriate amount of background information, succinct summary of research topic.

Methods: - Selection of Scenarios A-D are appropriate to the system, selection of fuel models seems appropriate, sensitivity analysis is informative.

Results: - Explained succinctly, figures are appealing and easy to read.

Discussion: - Most of the topics I had hoped would be addressed were addressed in the discussion, however, I think more detail could be provided about the evolutionary tradeoffs of scenescence corresponding both with hot weather and new growth, and how predicted changes in climate in the region may lead to that evolutionary coupling being maladaptive. For example, I could envision an additional analysis (or a future paper) that simulates predicted climate conditions for the four scenarios tested. - One topic that is only briefly addressed in the discussion is the limitations of the fire modeling behavior. I would like to see a more detailed discussion of these limitations, including whether or not the simulated behavior matches behavior observed in the field. If so, that would be additional compelling evidence for the authors’ conclusions.
